# Experimental and Analytical Analysis of Mechanical Properties for Large-Size Lattice Truss Panel Structure Including Role of Connected Structure

**DOI:** 10.3390/ma14175099

**Published:** 2021-09-06

**Authors:** Shaohua Li, Wenchun Jiang, Xiaolei Zhu, Xuefang Xie

**Affiliations:** 1State Key Laboratory of Heavy Oil Processing, College of New Energy, China University of Petroleum (East China), Qingdao 266580, China; joshuali_upc@126.com (S.L.); xiexuefang_upc@126.com (X.X.); 2School of Mechanical and Power Engineering, Nanjing Tech University, Nanjing 211816, China; zhuxiaolei@njtech.edu.cn

**Keywords:** large-size LTPS, mechanical properties, experimental and theoretical model, connected structure

## Abstract

The large-size lattice truss panel structure (LTPS) is continually increasing for higher upsizing, but the roles of its connected structures on the mechanical properties are always ignored during the previous structural integrity assessment. Thus, in this paper, a series of mechanical tests, including the fabricating of panel-to-panel LTPSs, monotonous tensile, and three- and four-point bending tests, were performed to comprehensively understand the mechanical behavior. Furthermore, a theoretical model including the role of connected structures was developed to predict both the elastic and plastic deformation behavior of panel-to-panel LTPS. Results show that the connected structure has a very significant effect on the mechanical properties of panel-to-panel LTPS during the three-bending tests, and I-beam element depresses its carrying capacity. The developed theoretical model was proved to accurately predict the experimental results, and the maximum error was limited within 20%. Finally, the dimensional effects of the connection components on mechanical properties were also analyzed by the theoretical model, and indicated that the panel-to-panel LTPS will present better mechanical performance than the intact structure when the width of I-beam element exceeds 12.2 mm or the its length downgrades to 39.1 mm, which provide a comprehensive guidance for the engineering design of large-size LTPS.

## 1. Introduction

The lattice truss panel structure (LTPS) [1,2] has a good application foreground in the automobiles, aerospace, and naval industries due to its superior comprehensive properties, including the high specific stiffness and strength, vibration damping, micro-wave absorption, and multifunction [3,4]. With the development of industry, the demand for the large-size LPTS is continually increasing for the higher upsizing [5,6]. However, at present, most existing researches are performed based on a representative element for simplicity [7], as the role of connected structures on the mechanical properties of the large-size LPTS is ignored. Hence, a new assessment theory of mechanical performance is required to guide its manufacturing and application.

At present, several methods have been developed to manufacture the PLTPS, including the investment casting (IC) [8,9], stamping-braze (SB) [10,11], additive manufacturing (AM) [12,13], and superplastic forming/diffusion bonding (SPF/DB) [14,15]. Nevertheless, they are seriously limited on the fabrication of large-size LPTS as a consequence of the difficulty of the reliably process (temperature, pressure, and vacuum) and the dimension of welding equipment [16,17]. Alternatively, due to the higher bonding strength, geometrical accuracy, lower condition requirement, and the ease to automation [18], the laser penetration welding (LPW) has been widely perceived as one of the most perspective method to fabricate large-size LTPS [19,20]. The fabrication process of large-size LTPS by the LPW is divided into a pre-manufacturing step to create single LTPSs and a subsequent welding step to manufacture the panel-to-panel LTPSs by a connected structure, as shown in Figure 1. Furthermore, the I-beam element is one of the most wildly-employed connected structure due to the excellent bearing capacity and flexible processing. A larger size of LTPS indicates a larger number of connected structures, and their roles on mechanical performance must be taken into consideration during the structural safety assessment. Unfortunately, most existing researches are still limited to characterize the mechanical behavior [21,22], bending performance [23], and failure modes [24,25] of the single LTPS, and the effect of connected structures is ignored. Pyszko [6] investigated the ultimate load-carrying capacity of the panel-to-panel joints by FEM, and the influence of geometric dimension of the connection component was analyzed. Niklas [26] proposed the searching for process of optimum geometry of a panel-to-panel joint of longitudinal arrangement. A configuration was searched for parameters which can ensure as-low-as-possible values of geometrical stress concentration coefficients at acceptable mass and deformations of the structure. Wang [27,28] and Ding [29] systematically studied the in-plane and out-of-plane strength of five types of connection structures. The results reveal that the cover plate joint and rectangular profile joint present more excellent combination properties, such as connection strength, construction technology, and structural weight, than other connection joints. Ehlers [30,31] designed several typical connection forms between the steel sandwich structure and the hull structure, and its relevant fatigue notch coefficient at the weld toe was calculated by the micro structural support theory. Then, the optimized design for the structural dimension was carried out by using this evaluation index. Recently, the mechanical properties of the sandwich panels in-plane connections were analyzed by FEM [32], and the design principles of new sandwich panels in-plane connections were raised. An innovative sandwich panels connection was proposed by Qiao [33], and the T-shaped and L-shaped connection constructions were tested under cyclic load. Wang [34] investigated the failure mode and hysteretic behavior of the composite beam connection, and a corresponding simulation model was proposed. However, the above researches were conducted by the FEM, and the experimental verification was still absent, with an aim to reveal the deformation mode, bearing capacity, ductility, degradation property, and fatigue life of the joint. The effect of the connected geometric dimensions on the mechanical properties of the panel-to-panel LTPS is ambiguous. In addition, a theoretical model, which is not only more convenient for engineering application but also reflects a deeper understanding of deformation mechanism, is also still required to predict the overall deformation behavior of the panel-to-panel construction.

Therefore, in this study, the mechanical performance of the LTPS with panel-to-panel construction was investigated by the experimental and theoretical methods. The three-point and four-point bending mechanical tests were carried out for both the I-beam and intact panel-to-panel LTPSs to comprehensively reveal the role of connected structure. Additionally, a theoretical model was developed to accurately predict both the elastic and plastic deformation behavior. Finally, the dimensional effects of the I-beam (width *d* and length *w*) on mechanical properties were also analyzed to provide a comprehensive guidance for the engineering design of large-size LTPS.

## 2. Experimental

### 2.1. Materials Characterization

Owing to the combination of excellent proprieties, including the mechanical properties processability and weldability, the Q345 steel was favored as the engineering structure material in different industries. Hence, the Q345 steel was used to fabricate the LTPSs in this study, and its chemical compositions are listed in Table 1, which was solution-treated at 1070–1100 °C. To fully understand the mechanical properties of the base material and butt joint, the monotonic tensile tests were performed on a MTS servo-hydraulic machine (MTS Inc., Huntsville, AL, USA) at ambient temperature. The CO_2_ gas shielded welding was employed in this study, whose welding current, voltage, and speed are 150 A, 23 V, and 1.3 mm/s, respectively. The samples were cut with a parallel length of 240 mm and a fillet of 25 mm by a high-pressure water jet, as exhibited in Figure 2a. The upper chuck was fastened. However, the lower chuck was imposed on a constant displacement with speed of 2.0 mm/min until the specimen fracture. The deformation of the parallel section was monitored and controlled by an axial extensometer with a clip gauge of 20 mm, and the magnitudes of axial loads imposed to the specimen were measured by a force sensor. To improve the accuracy, two groups of repeated tests were conducted.

The stress-strain responses of the base material and butt joint are plotted in Figure 2b, and the measured elastic modulus, yield strength, and ultimate strength are listed in Table 2. The primary elastic module, subsequent plasticity, and final rupture were found for two materials. However, for the welded samples, the fracture produces at the parallel section instead of the joints, which implies that the joint presents higher mechanical strength and the welding quality is reliable enough.

### 2.2. Fabricating Process of Panel-to-Panel LTPSs

The fabricating process of panel-to-panel LTPS is illustrated in Figure 3. Firstly, the lattice core and metal sheets were cut by the high-pressure water jet (Figure 3a). The water column was set as deviating from the materials by 0.1 mm, and the interlocking slots were cut at the central-axis of the unit-cells. Its height and width were equal to the thickness of metal panel. Secondly, the lattice cores and metal sheets were polished, cleaned and dried in order, and the inner cores were finished by connecting the horizontal and vertical lattice cores through the interlocking-slots (Figure 3b). Then, the single LTPS was assembled by the two metal sheets and lattice core (Figure 3c), which were then bonded by the LPW process with laser power; the welding speed and separation of 5000 W were 1.77 m/min and 0 mm, respectively (Figure 3d). Finally, the panel-to-panel LTPSs were manufactured by connecting the single LTPSs through an I-beam element (Figure 3e). It should be noted that, to better illustrate the role of connected structure, the intact LTPSs with same size were also manufactured.

### 2.3. Mechanical Tests

The mechanical properties of both I-beam and intact panel-to-panel LTPSs were characterized by the three-point and four-point bending tests on a MTS electro-hydraulic machine (1200 kN) in accordance with the standard ASMT D7249-18, as show in Figure 4. Note that the lower rollers were fastened, and the upper roller was imposed on the location of I-beam connected structure during the three-point bending tests (Figure 5a) while they were located at the bilateral single LTPSs for the four-point bending tests (Figure 5b), which is beneficial to comprehensively understand the role of connected structure. In additional, during the tests, the displacement of upper roller was controlled as a consistent rate of 2.0 mm/min. The deformation of the structure was monitored by an axial displacement extensometer with a gauge of 25 mm. The loads imposed on the specimen were measured by a force sensor. The imposed load was recorded to calculate its flexural rigidity and ultimate load of the panel-to-panel constructions. The flexural rigidity in the three-point (*D*_3_) and four-point (*D*_4_) bending tests was calculated by Equations (1) and (2), respectively.
(1)$$D_{3}=\frac{\left(P_{1}-P_{2}\right)L^{3}}{\Delta \delta }$$
(2)$$D_{4}=\frac{\left(L-S\right)H_{c}}{4}\frac{P_{1}-P_{2}}{\left(\varepsilon_{1}-\varepsilon_{2}\right)+\left(\varepsilon_{1}^{\prime }-\varepsilon_{2}^{\prime }\right)}$$
where Δ*δ* is the increment of the deflection. *L*, *S*, and *H*_c_ denote the supporting span, loading span and structure height, respectively; $\varepsilon_{1}$ and $\varepsilon_{2}$ are the strain at initial elastic, and $\varepsilon_{1}^{\prime }$ are $\varepsilon_{2}^{\prime }$ are the strain at final elastic stage; $P_{1}$ and $P_{2}$ represent the loading measured from the upper metal sheet when the strain reach to $\varepsilon_{1}$ and $\varepsilon_{2}$, respectively. In this study, the height and width of the single structure were 50 mm and 150 mm, respectively. The supporting spans for three-point bending tests and four-point bending tests were 300 mm and 1000 mm, and the loading spans for four-point bending tests were 300 mm.

## 3. Analytical

The key dimensions of the LPTSs involved in the analytical model are firstly introduced. The construction consists of two single sandwich structures and an I-beam structure, and the height and width of the single structure are *H*_c_ + 2*t* and *B*, respectively, as shown in Figure 5a. The schematic of the unit cell and I-beam element were illustrated in Figure 6a,b. The width (*d*) is equal to the thickness (*t*) of the metal sheet, and the length *w* of the frange-plate is equal to the length (*L*_c_) of the unit cell. The schematic of the typical unit cell and I-beam are illustrated in Figure 6a, whereby number 1,2,3 and 4 present the initial individual trusses, and the *l_x_*, *l*_y_ and *l*_z_ are the dimension of the unit cell in *X*, *Y* and *Z* direction, respectively.

Figure 7a,b describe the loading distribution of the individual truss under shearing and compression, respectively. Under the shearing load, the metal sheets and platforms provide no contribution to the stiffness or strength of the LTPS. Hence, the longitudinal force *N* and tangential force *Q* on the truss 1 and 2 can be obtained by Equations (3) and (4), respectively,
(3)$$N=F\sin\theta -F_{R}\cos\theta$$
(4)$$Q=F\cos\theta +F_{R}\sin\theta$$
where *F* and *F*_R_ are loaded force and counter-force at the truss tip, respectively. *θ* denotes the inclination angle of the truss.

The equilibrium equation of bending moment *M* at truss end is
(5)$$2M=\left(F\cos\theta +F_{R}\sin\theta \right)l$$

In the horizontal direction, both the displacement and rotation angle of the truss are zero due to the clamped supporting.

Deflection:(6)$$\frac{-F\cos\theta +F_{R}\sin\theta }{t^{2}}l\sin\theta +\left(-\frac{6Ml^{2}}{Et^{4}}+2\frac{F\sin\theta +F_{R}\cos\theta }{Et^{4}}l^{3}\right)\cos\theta =0$$

Rotation angle:(7)$$\frac{12Ml}{Et^{4}}-6\frac{P\sin\theta +P_{R}\cos\theta }{Et^{4}}l^{2}=0$$

The corresponding longitudinal force, tangential force, and moment on the truss 1 and 2 can be written as
(8)$$N=F\frac{t^{2}l^{2}\cos\theta }{t^{4}\sin^{2}\theta +t^{2}l^{2}\cos\theta }$$
(9)$$Q=F\frac{t^{2}l^{2}\sin\theta }{t^{4}\sin^{2}\theta +t^{2}l^{2}\cos\theta }$$
(10)$$M=Fl\frac{6t^{4}\sin\theta }{t^{4}\sin^{2}\theta +t^{2}l^{2}\cos^{2}\theta }$$

Since the total displacement is the superposition of deformation controlled by longitudinal force, tangential force, and bending moment, the deformation contribution in X direction can be calculated by the dummy-load method:(11)$$\delta_{x}=\frac{-F\cos\theta +F_{R}\sin\theta }{t^{2}}l\cos\theta +\left(\frac{6Ml^{2}}{Et^{4}}-2\frac{F\sin\theta +F_{R}\cos\theta }{Et^{4}}l^{3}\right)\sin\theta =\frac{Fl^{3}}{E\left(t^{4}\sin^{2}\theta +t^{2}l^{2}\cos\theta \right)}$$

Then, the shearing force *T*_1,2_ imposed on the truss tip is expressed as
(12)$$T_{1,2}=\frac{EA_{}^{c}}{l_{x}}\delta_{x}\sin^{2}\theta +\frac{12EI}{{\left(l_{x}\right)}^{3}}\delta_{x}\cos^{2}\theta$$
where *A*^c^ is the area of the unit cell. However, it is worth noting that the deformation of the truss 3 (or 4) are dominated by the tangential deformation, and that the longitudinal deformation remains zero. The corresponding longitudinal force, tangential force, and bending moment at truss tip are written as Equations (13)–(15).
(13)N=0
(14)$$Q=\frac{12EI}{{\left(l_{x}\right)}^{3}}\delta_{x}$$
(15)$$M=\frac{6EI}{{\left(l_{x}\right)}^{2}}\delta_{x}$$

Then, the equivalent force *T*_3,4_ imposed for the 3 (or 4) truss is:(16)$$T_{3,4}=Q=\frac{12EI}{{\left(l_{x}\right)}^{3}}$$

The relation between force and deformation can be expressed as
(17)$$T_{xy}=2\left(EA_{}^{c}\sin^{2}\theta +\frac{12EI}{{\left(l_{x}\right)}^{2}}\cos^{2}\theta +\frac{12EI}{{\left(l_{x}\right)}^{2}}\right)\gamma^{c}\sin\theta$$
where *T_xy_* and *γ*^c^ are the shearing force and strain of the unit cell. Finally, the equivalent shearling modulus of a unit cell is obtained.
(18)$$G_{}^{c}=\frac{\tau^{c}}{\gamma^{c}}=\frac{2E}{A_{}^{c}}\left(A_{}^{c}\sin^{2}\theta +\frac{12I}{{\left(l_{x}\right)}^{2}}\cos^{2}\theta +\frac{12I}{{\left(l_{x}\right)}^{2}}\right)\sin\theta$$
where *G*^c^ and *τ*^c^ are the shearing modulus and stress of the unit cell.

The shearing deformation of the panel-to-panel construction is the superposition of the shearing deformation of the single LTPS and I-beam. In addition, based on the equivalent homogenization theory, the equivalent shearing strain is written as:(19)$$\gamma^{*}=\frac{\tau^{*}}{{\left[G\right]}^{*}}=\frac{2F}{\left(A^{c}+A^{I}\right){\left[G\right]}^{*}}$$
(20)$$\gamma^{*}=\gamma^{c}+\gamma^{I}=\frac{F}{A^{c}G_{}^{c}}+\frac{F}{A^{I}G^{I}}$$
where *γ*^I^ and *γ** are the shearing strain of the I-beam and whole structure. *τ** and [*G*]* are the shearing stress and modulus of the whole structure, respectively. *A*^I^ denotes the area of the I-beam element.

The equivalent shearing modulus of the panel-to-panel construction is
(21)$${\left[G\right]}^{*}=\frac{2A^{c}A^{I}G_{}^{c}G^{I}}{\left(A^{c}+A^{I}\right)\left(A^{c}G_{}^{c}+A^{I}G^{I}\right)}$$
(22)$$G^{I}=\frac{d}{w}G$$
(23)$${\left[G\right]}^{*}=\frac{2\left(L-w\right)dG_{}^{c}G}{L\left[\left(L-w\right)G_{}^{c}+dG\right]}$$
where *G*^I^ and *G* are the shearing modulus of the I-beam and material.

Based on this method, the equivalent elastic modulus *E*^c^ of the single LTPS in *Z* direction is derived.
(24)$$E^{c}=\frac{\left(t^{4}\sin^{2}\theta +t^{2}l^{2}\cos^{2}\theta \right)\cos\theta }{{\left(2l\sin\theta +b+c\right)}^{2}l^{2}}E$$
where *E* is the elastic modulus of the material.

The overall compressive force of the panel-to-panel construction is the superposition of the force shared on the of the single LTPS and I-beam. Besides, the overall compressive strain is equal to the strain of the single LTPS, which is also equal to the compressive strain of the I-beam, thus
(25)$$\varepsilon^{*}=\frac{\sigma^{*}}{{\left[E\right]}^{*}}=\frac{\sigma^{c}A^{c}+\sigma^{I}A^{I}}{\left(A^{c}+A^{I}\right){\left[E\right]}^{*}}$$
(26)$$\varepsilon^{*}=\frac{\sigma^{c}}{E^{c}}=\frac{\sigma^{I}}{E^{I}}$$
where *ε** denotes the equivalent compressive strain of the unit cell. In addition, [*E*]* and *E*^I^ are the equivalent elastic modulus of the whole structure and I-beam element. *σ**, *σ*^c^, and *σ*^I^ are the equivalent compressive stress on the whole structure, unit cell, and I-beam element, respectively. The equivalent elastic modulus of the panel-to-panel construction is
(27)$${\left[E\right]}^{*}=\frac{E^{c}A^{c}+E^{I}A^{I}}{\left(A^{c}+A^{I}\right)}$$
(28)$$E^{I}=\frac{d}{w}E$$
(29)$${\left[E\right]}^{*}=\frac{E^{c}\left(L-w\right)+E^{I}w}{L}$$

Under the three-point bending load, the total deformation of the panel-to-panel construction contains of the bending deformation from metal sheet and the shearing deformation from lattice cores, as shown in Figure 8. The relationship between bending moment and curvature for sheet metal is
(30)$$\frac{M}{EI}=\left(-\frac{1}{\rho }\right)$$
where *I* and *ρ* denote the moment of inertia and radius of curvature.

The shearing deformation sketch of inner core is shown in Figure 9, and the corresponding relation between shearing stress and strain is
(31)$$\tau =\frac{EQt\left(L_{z}+t\right)}{2D}=\frac{Q}{B\left(L_{z}+t\right)}$$
(32)$$\gamma =\frac{Q}{{\left[G\right]}^{*}B\left(L_{z}+t\right)}$$
where *D* is the flexural rigidity of the inner core. *τ* and *γ* are the shearing stress and strain of the inner core, respectively. According to the stress-strain relation, the balance equation is written as
(33)$$\frac{d\Delta^{c}}{dx}=\frac{QL_{z}}{{\left[G\right]}^{*}B{\left(L_{z}+t\right)}^{2}}$$
(34)$$\Delta^{c}=\frac{WLL_{z}}{4{\left[G\right]}^{*}B{\left(L_{z}+t\right)}^{2}}$$
where Δ^c^ is the deformation of the inner core.

Since the total central deflection is the superimposition of the bending deformation of the sheet metals and the shearing deformation of the lattice core, the deflection in Z direction is:(35)$$\Delta =\Delta^{f}+\Delta^{c}=\frac{WL^{3}}{24EBt{\left(t+L_{z}\right)}^{2}+12EBt^{3}+12{\left[E\right]}^{*}B{\left(L_{z}\right)}^{3}}+\frac{WLL_{z}}{4{\left[G\right]}^{*}B{\left(L_{z}+t\right)}^{2}}$$
where Δ^f^ is the deformation of the inner core.

Based on the experimental results [21,23], about 5% percentage deformation is account by elastic, and the plastic behaviors of the panel-to-panel construction will be discussed in next. The plastic deformation modes of the truss under shearing and compressive stress are shown in Figure 10a,b, respectively. Δ*_x_* and Δ*_z_* denote the total deformation in *X* and *Z* direction. Furthermore, 1′ (2′, 3′, and 4′) represent the deformed trusses. The corresponding rotation angle of the plastic hinge is *α*_1,1_.

Under shearing force, the rotation angles of the plastic hinge for the truss 1, 3, and 4 are easily calculated by Equations (32) and (33).
(36)$$\alpha_{1,1}=2\left[\theta -\arcsin\frac{l_{z}}{\sqrt{{\left(l_{z}\right)}^{2}+{\left(l_{x}+\Delta_{x}\right)}^{2}}}\right]$$
(37)$$\alpha_{3,1}=\alpha_{4,1}=2\arcsin\frac{\Delta_{x}}{l}$$
where *α*_1,1_, *α*_3,1_, and *α*_4,1_ are the plastic rotation angle of the truss 1, 3, and 4, respectively. Δ*_x_* denotes the displacement of the truss in X direction.

However, for the truss 2, the balance relation between rotation angle and deformation can be expressed as:(38)$$\left\{ \begin{array}{l} \frac{l}{2}\sin\left(\beta -\alpha_{2,1}\right)+\frac{l}{2}\cos\left(\theta -\alpha_{2,2}\right)=l_{x}-\Delta_{x}\\ \frac{l}{2}\cos\left(\beta -\alpha_{2,1}\right)+\frac{l}{2}\sin\left(\theta -\alpha_{2,2}\right)=l_{z} \end{array}\right.$$

The rotation angle of the truss 2 is obtained by solving Equation (34).
(39)$$\alpha_{2}=\sum_{n=1}^{4}\alpha_{2,n}=2\arccos\frac{{\left(l_{x}-\Delta_{x}\right)}^{2}+{\left(l_{z}\right)}^{2}}{l^{2}}$$
where *α*_2_ the plastic rotation angle of the truss 2.

According to the energy balance between work from the loading force and the absorption energy from the structural deformation, the relation between applied shearing force *F_x_* and shearing deformation Δ*_x_* is calculated by Equation (40).
(40)$$\int_{0}^{\Delta_{x}}F_{x}dx=\sigma_{y}t^{2}\Delta l+M\sum_{n=1}^{4}\alpha_{n}$$
where $\Delta l=\sqrt{{\left(l_{x}-\Delta_{x}\right)}^{2}+{\left(l_{z}\right)}^{2}}-l$, *σ_y_* is the yield strength of the base material. Δ*l* denotes the decrement of the truss under shearing load. By differentiating Equation (40), the relation between shearing force *F_x_* and shearing deformation Δ*_x_* can be expressed as:(41)$$F_{x}=\frac{\sigma_{y}t^{2}\left(l_{x}+\Delta_{x}\right)}{\sqrt{{\left(l_{z}\right)}^{2}+{\left(l_{x}+\Delta_{x}\right)}^{2}}}+\frac{\sigma_{y}t^{3}}{4}\left[\frac{2l_{z}}{{\left(l_{z}\right)}^{2}+{\left(l_{x}+\Delta_{x}\right)}^{2}}+\frac{4}{\sqrt{l^{2}+{\left(\Delta_{x}\right)}^{2}}}+\frac{4\left(l_{x}-\Delta_{x}\right)}{\sqrt{{\left(l_{x}\right)}^{2}-{\left(l_{x}-\Delta_{x}\right)}^{2}}\sqrt{{\left(l_{z}\right)}^{2}+{\left(l_{x}-\Delta_{x}\right)}^{2}}}\right]$$

Similarly, for compressive condition, the rotation angle of the plastic hinge and energy balance equation is expressed as:(42)$$\alpha =16\Delta \arccos\frac{\sqrt{{\left(l\sin\theta \right)}^{2}+{\left(l\cos\theta -\Delta_{z}\right)}^{2}}}{l}$$
(43)$$\int_{0}^{\Delta_{z}}F_{z}dz=M\Delta \alpha$$
where Δ*_z_* and *F_z_* are the compressive deformation and force in Z direction. *F_z_*. *α* is the plastic rotation angle. By differentiating Equation (43), the equivalent compressive force in *Z* direction is
(44)$$F_{z}=\frac{4\left(l_{z}-\Delta_{z}\right)}{\sqrt{{\left(l_{z}\right)}^{2}-{\left(l_{z}-\Delta_{z}\right)}^{2}}\sqrt{{\left(l_{x}\right)}^{2}+{\left(l_{z}-\Delta_{z}\right)}^{2}}}\sigma_{y}t^{3}$$

For the whole structure, an equilibrium requires that the work carried by the load is related to the bended sheet metals, shearing cores and compressive cores.
(45)$$\int_{0}^{\Delta_{z}}Wdz=4M\arctan\frac{2\Delta_{z}}{L}+\int_{0}^{\Delta_{x}}F_{x}dx+\int_{0}^{\Delta_{z}}F_{z}dz$$

By differentiating Equation (45), the relation between total force *W* and overall deformation Δ*_z_* can be expressed as Equation (46).
(46)$$W=F^{f}+F_{x}+F_{z}$$
where $F^{f}=\frac{2LBt^{2}\sigma_{ys}}{L^{2}+4{\left(\Delta_{z}\right)}^{2}}$

$$F_{x}=\frac{\sigma_{y}t^{2}\left(l_{x}+\frac{2L_{z}}{L}\Delta_{z}\right)}{\sqrt{{\left(l_{z}\right)}^{2}+{\left(l_{x}+\frac{2L_{z}}{L}\Delta_{z}\right)}^{2}}}+\frac{\sigma_{y}t^{3}}{4}\left[\frac{2l_{z}}{{\left(l_{z}\right)}^{2}+{\left(l_{x}+\frac{2L_{z}}{L}\Delta_{z}\right)}^{2}}+\frac{4}{\sqrt{l^{2}+{\left(\frac{2L_{z}}{L}\Delta_{z}\right)}^{2}}}+\frac{4\left(l_{x}-\frac{2L_{z}}{L}\Delta_{z}\right)}{\sqrt{{\left(l_{x}\right)}^{2}-{\left(l_{x}-\frac{2L_{z}}{L}\Delta_{z}\right)}^{2}}\sqrt{{\left(l_{z}\right)}^{2}+{\left(l_{x}-\frac{2L_{z}}{L}\Delta_{z}\right)}^{2}}}\right]$$

$$F_{z}=\frac{4\left(l_{z}-\frac{l_{x}\Delta_{z}}{L}\right)}{\sqrt{{\left(l_{z}\right)}^{2}-{\left(l_{z}-\Delta h\right)}^{2}}\sqrt{{\left(l_{x}\right)}^{2}+{\left(l_{z}-\frac{l_{x}\Delta_{z}}{L}\right)}^{2}}}\sigma_{y}t^{3}$$

For four-point bending, the decomposed forces can be expressed as
(47)$$F^{f}=\frac{2\left(L-S\right)Bt^{2}\sigma_{y}}{{\left(L-S\right)}^{2}+4{\left(\Delta_{z}\right)}^{2}}$$
(48)$$F_{x}=\frac{\sigma_{y}t^{2}\left(l_{x}+\frac{2L_{z}}{\left(L-S\right)}\Delta_{z}\right)}{\sqrt{{\left(l_{z}\right)}^{2}+{\left(l_{x}+\frac{2L_{z}}{\left(L-S\right)}\Delta_{z}\right)}^{2}}}+\frac{\sigma_{y}t^{3}}{4}\left[\frac{2l_{z}}{{\left(l_{z}\right)}^{2}+{\left(l_{x}+\frac{2L_{z}}{\left(L-S\right)}\Delta_{z}\right)}^{2}}+\frac{4}{\sqrt{l^{2}+{\left(\frac{2L_{z}}{\left(L-S\right)}\Delta_{z}\right)}^{2}}}+\frac{4\left(l_{x}-\frac{2L_{z}}{\left(L-S\right)}\Delta_{z}\right)}{\sqrt{{\left(l_{x}\right)}^{2}-{\left(l_{x}-\frac{2L_{z}}{\left(L-S\right)}\Delta_{z}\right)}^{2}}\sqrt{{\left(l_{z}\right)}^{2}+{\left(l_{x}-\frac{2L_{z}}{\left(L-S\right)}\Delta_{z}\right)}^{2}}}\right]$$
(49)$$F_{z}=\frac{4\left(l_{z}-\frac{l_{x}\Delta_{z}}{\left(L-S\right)}\right)}{\sqrt{{\left(l_{z}\right)}^{2}-{\left(l_{z}-\Delta h\right)}^{2}}\sqrt{{\left(l_{x}\right)}^{2}+{\left(l_{z}-\frac{l_{x}\Delta_{z}}{\left(L-S\right)}\right)}^{2}}}\sigma_{y}t^{3}$$

## 4. Results and Discussion

The experimental load–displacement curves for both intact and I-beam panel-to-panel construction under three-point and four-point bending tests are plotted in Figure 11a,b, respectively. It is obvious that, regardless of the three-point and four-point bending, the mechanical responses can be divided into three stages. The load versus displacement response was characterized by an initial elastic regime (stage I) when the overall deformation was dominant by the elastic bending deformation of the metal sheets and coerced shearing deformation of the inner core. In case of three-point bending test, the flexural rigidities of the intact and I-beam element structure were 1.72 × 10^8^ N·mm^2^ and 1.37 × 10^8^ N·mm^2^, and the corresponding initial yield loads were 68.54 kN and 53.90 kN, respectively. Then, the slope decreased significantly with the increase in imposed displacement after the plastic bending deformation became aroused by the contact between the upper metal sheet and the indenter, and the displacement loading reaches the inflection point (stage II). The bending deformation of the metal sheets were significantly larger than the shearing deformation of the inner core. It implies that the metal sheets yielded earlier than the inner core. The plastic collapse loads of the I-beam element and intact structure were 80.15 kN and 82.29 kN, respectively, and the corresponding ratio was 97.40%.

After a peak, the load decreased gradually with the further increase in displacement due to the debonding between metal sheet and lattice core (stage III) Similar characteristics were observed in four-point bending tests until the peak, and the major inflections and nonlinearities of intact and I-beam element structure were initiated at similar displacement, indicating the similarity of underlying damage mechanisms in that region. The difference of flexural rigidity between intact and I-beam element structure was 0.52 × 10^8^ N·mm^2^, and the corresponding yield loads were 46.58 kN and 43.63 kN, respectively. However, the curves elevate unceasingly with a smaller rate due to the yield of the inner core. The measured plastic collapse loads of the intact and I-beam element were 68.58 kN and 65.53 kN. Appreciable differences arose with respect to the extent of the load drops and fluctuations near the peak values which corresponded to the dominant failure mode, i.e., debonding. Fortunately, the butted joints were intact after the three-point bending and four-point bending tests.

As a conclusion, the butted joints present good mechanical performance, and the overall deformation is the superimposition of the bending deformation of metal sheets, and the coerced shearing deformation of inner core. Elastic deformation of the metal sheets and inner core is defined in stage I. The overall deformation in stage II is dominant by the plastic deformation of the sheet metals and elastic deformation of the inner core. Both the metal sheets and inner core yield in stage III and the debonding failure reduces the bearing capacity of those structure further. For the investigated dimension, the I-beam element structure have slight poorer performance due to the weakening effect of the I-beam. Furthermore, the comparison of mechanical response for two structures is worth more attention.

For both three- and four-point bending, the yield load is determined by equating the maximum bending moment within the structure to the plastic collapse moment of the section. For example, in case of three-point bending, the maximum bending moment *M*_m_ is located at mid of the metal sheet, and the plastic deformation occurs when the metal sheets attain the yield strength, giving
(50)$$\frac{M_{m}}{Bt_{f}H_{c}}\leq \sigma_{y}^{}$$
(51)$$M_{\max}=\frac{WL}{4}$$

Hence, the critical load (*W*^f^) of the metal sheet is
(52)$$W^{f}\leq \frac{4Bt_{f}H_{c}}{L}\sigma_{y}^{}$$

In addition, the inner core shears plastic when the panel is subjected to a transverse shear force. A simple work balance gives the collapse load, assuming that the face sheets on the right half of the sandwich panel rotate through an angle *φ*, and that those on the left half rotate through an angle *φ*. Consequently, the inner core shears by an angle *φ*. On equating the external work carried out with *Wlφ*/2 to the internal work dissipated within the core of length and at the two plastic hinges in the face sheets, we obtain
(53)$$W_{}^{f}\Delta d_{△}=4M\Delta d\varphi$$
(54)$$W_{}^{c}\Delta d_{△}=2BH_{c}\tau_{y}^{c}$$
where *W*^c^ and *τ^c^_y_* are the collapse load and shearing yield strength of the inner core, respectively.

Hence, the critical load [*W*^c^]* of the inner core is
(55)$${\left[W^{c}\right]}^{*}\leq W_{}^{f}+W_{}^{c}=\frac{2B{\left(t_{f}\right)}^{2}}{L}\sigma_{y}^{}+2BH_{c}\tau_{y}^{c}$$

Similarity, under four-point bending, the critical load of the metal sheet and the inner core collapse are expressed as Equations (56) and (57), respectively.
(56)$$W^{f}\leq \frac{4Bt_{f}H_{c}}{L-S}\sigma_{y}^{}$$
(57)$${\left[W^{c}\right]}^{*}\leq W_{}^{f}+W_{}^{c}=\frac{2B{\left(t_{f}\right)}^{2}}{L-S}\sigma_{y}^{}+2BH_{c}\tau_{y}^{}$$

To summarize, the theoretical load-displacement curve was divided at the third stage.

Stage I: loading range $0\leq W\leq W^{f}$

$$W=\frac{\Delta_{z}^{}}{\frac{L^{3}}{24EBt{\left(t+L_{z}\right)}^{2}+12EBt^{3}+12{\left[E\right]}^{*}B{\left(L_{z}\right)}^{3}}+\frac{LL_{z}}{4{\left[G\right]}^{*}B{\left(L_{z}+t\right)}^{2}}}$$

Stage II: loading range
$W^{f}\leq W\leq {\left[W^{c}\right]}^{*}$

$$W=W^{f}+F^{f}+\frac{4{\left[G\right]}^{*}B{\left(L_{z}+t\right)}^{2}}{LL_{z}}\Delta_{z}^{}$$

Stage III: loading range $W\geq {\left[W^{c}\right]}^{*}$

$$W={\left[W^{c}\right]}^{*}+F^{f}+F_{x}+F_{z}$$

Based on the above analytical models, the theoretical load-displacement curves are plotted in Figure 12. In the elastic, initial yield and the plastic collapse stage, the theoretical model has a good agreement with the experimental result, and the relatively great error is only found in the failure stage. The deformation in stage I contains the elastic bending deformation of the metal sheets and shearing deformation of the inner cores. The predicted flexural rigidities of the intact and I-beam element structure are 1.54 × 10^8^ N·mm^2^ and 1.52 × 10^8^ N·mm^2^, respectively, and the corresponding error between analytical model and experimental are 10.47% and 10.95%, respectively. However, in the second stage, the deformation is the superimposition of the plastic deformation of the metal sheets and the elastic deformation of the inner core. The plastic deformation of the inner core is behind of the plastic deformation of the metal sheets because the shearing deformation Δ*_x_* (=$\frac{l_{x}\Delta_{z}}{L-S}$) is far less than the total displacement increment Δ*_z_*. The predicted initial yield loads are 70.11 kN and 64.02 kN, and the corresponding errors between analytical model and experimental are 2.29% and 18.78%, respectively. In stage III, the total deformation is dominant by the plastic deformation both of the metal sheet and inner core. The predicted plastic collapse loads of the intact and I-beam element structure are 84.66 kN and 80.05 kN, and the corresponding error between analytical model and experimental is smaller than 5%, respectively. Theoretically, the load elevates slightly with the increase in the displacement. Nevertheless, the error between analytical model and experiment increases gradually. This is because the high-level stress raises in the bonding section, caused by deformation disharmony between the metal sheets and inner core. The debonding occurs when the localized stress reaches ultimate strength of the laser welding joint, and the structural bearing capacity decreases sharply. The maximum error of the predicted and experimental result is 10.31% for four-point bending. It is worth noting that, in the case of three-point bending, failure occurs earlier than that under four-point bending as a result of the greater deformation disharmony of the metal sheet and inner core.

The effects of I-beam width (*d*) and length (*w*) on the mechanical properties of the panel-to-panel construction are exhibited in Figure 13. Considering the width range from 3 mm to 15 mm, i.e., the normalized dimension range from 0.3 to 3, the flexural rigidity, initial yield load, and plastic collapse load increase 22.08%, 9.23%, and 7.50%, respectively. It implies that width has no significant influence on the initial yield load and plastic collapse load, regardless of the flexural rigidity in the investigated range. Theoretically, the flexural rigidity increases with the increase in the I-beam width as a result of the elevated equivalent elastic modulus and shearing modulus of the panel-to-panel LTPS. However, the effective length of the metal sheet decreases slightly with the increase in the I-beam width. According to the failure criteria, the I-beam width has smaller influence on the yield and collapse of metal sheet and inner core. The flexural rigidity, initial yield load, and plastic collapse load decrease significantly with the elevation of the beam length; its decrement are 85.16%, 67.48%, and 67.69%, respectively. The equivalent elastic modulus and equivalent shearing modulus of the panel-to-panel construction decreases with the elevation of the length. As a result, the flexural rigidity decreases significantly. The effective length of the metal sheet increases with the increase in the I-beam length. Based on the failure criteria, both the initial yield load and plastic collapse load decreases. Under same normalized dimension, the decreasing amplitude stimulated by the beam length greatly exceed the increment controlled by beam width. The mechanical properties of the panel-to-panel construction are more sensitive to the beam length. In addition, the analytical models conclude that the I-beam element connected to panel-to-panel LTPS presents better mechanical performance than the intact structure when the I-beam width exceeds 12.2 mm or the I-beam length downgrades to 39.1 mm.

It is worth noting that the deformation of the panel-to-panel LTPS can transform to mode Ⅱ with the increase in I-beam length, as shown in shown in Figure 14. For this deformation mode, the equilibrium relation under three-point and four-point bending was written as Equation (58).
(58)$$\int_{0}^{\Delta_{z}}Wdz=8M_{p}\arccos\frac{2\Delta_{z}}{L}+\int_{0}^{\Delta_{x}}F_{x}dx+\int_{0}^{\Delta_{z}}F_{z}dz$$

By differentiating Equation (58), the force on the sheet metal under three-point and four-point bending are Equations (59) and (60), respectively.
(59)$$F^{f}=\frac{4LBt^{2}\sigma_{ys}}{L^{2}+4{\left(\Delta_{z}\right)}^{2}}$$
(60)$$F^{f}=\frac{4Bt^{2}\sigma_{ys}}{{\left(L-S\right)}^{2}+4{\left(\Delta_{z}\right)}^{2}}$$

## 5. Conclusions

In this paper, the mechanical properties of the panel-to-panel LTPSs were investigated by the experimental and theoretical methods. The roles of connected structure were revealed to provide an extensive guidance for the engineering application and design of the large-size LTPS. The main conclusions are shown as the following:

Under three-point bending, the connected structure had a very significant effect on the mechanical properties of panel-to-panel LTPS, and I-beam element greatly depressed its carrying capacity. The degradation of the flexural rigidity, initial yield load, and plastic collapse load are 20.59%, 21.38%, and 1.30%, respectively. Nevertheless, this effect became inconspicuous (less than 5%) under four-point bending, whereby the external loading was far from the I-beam element. This implies that the external loading being set at the location away from a connected structure is a positive for engineering LTPS in order to attain a greater carrying capacity.

(1)The analytical models, including flexural rigidity, initial yield, and plastic collapse, were proposed. The maximum error is 18.78%, which reveals that the models were proven to accurately predict the deformation behavior and provide more convenience for the engineering safety assessment.(2)The dimensional effects of the connection components on mechanical properties were discussed by the analysis models. The mechanical properties were enhanced by elevating the I-beam width *d* and decreasing the I-beam length *w*, which are more sensitive to the length. In addition, those models provide a guidance for the engineering design of large-size LTPS. The I-beam element connected panel-to-panel LTPS presents better mechanical performance than the intact structure when the *d* exceeds 12.2 mm or the *w* downgrades to 39.1 mm.

## Figures and Tables

**Figure 1 materials-14-05099-f001:**
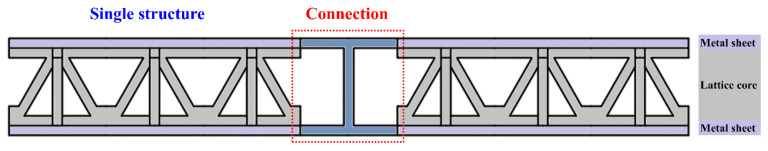
Schematic of panel-to-panel construction.

**Figure 2 materials-14-05099-f002:**
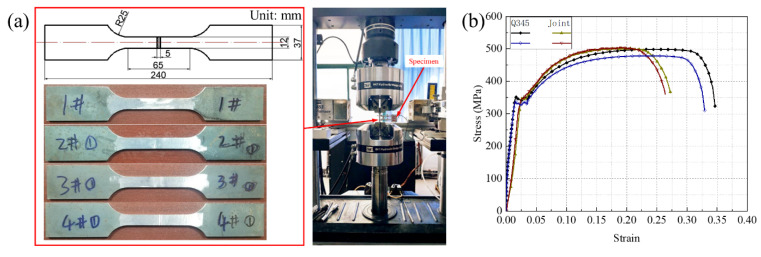
Detailed experimental setup (**a**) and stress-strain curves (**b**) for base material and butt joint.

**Figure 3 materials-14-05099-f003:**
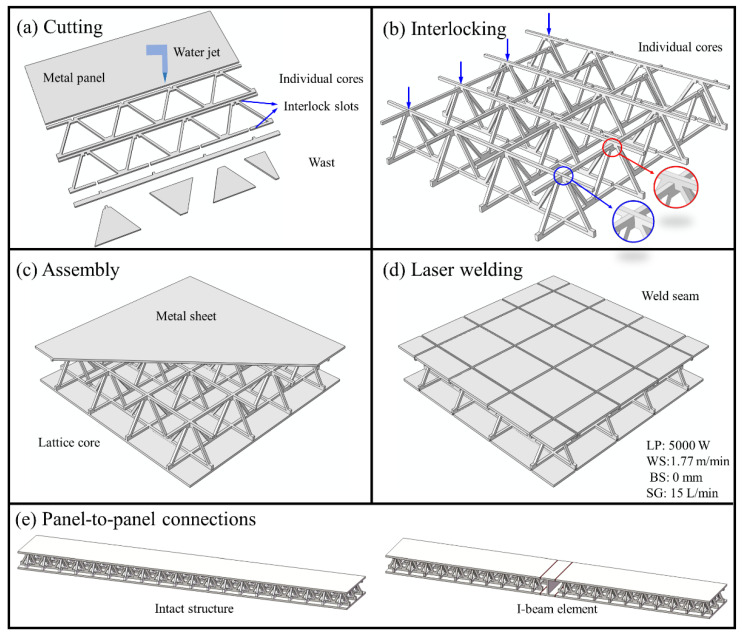
Fabricating process of panel-to-panel LTPSs: (**a**) cutting metal sheets and individual cores, (**b**) individual cores connected by interlock, (**c**) assembly inner cores and metal sheet, (**d**) connection by laser welding, (**e**) manufactured large-size LTPSs: intact structure and I-beam element connected structure.

**Figure 4 materials-14-05099-f004:**
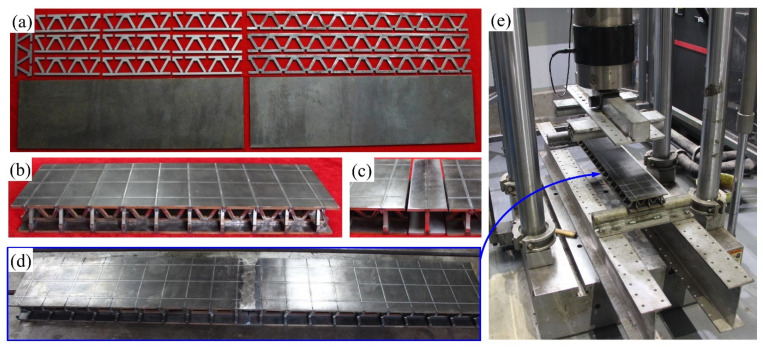
Detailed manufactured samples (**a**–**d**) and experimental setup (**e**).

**Figure 5 materials-14-05099-f005:**
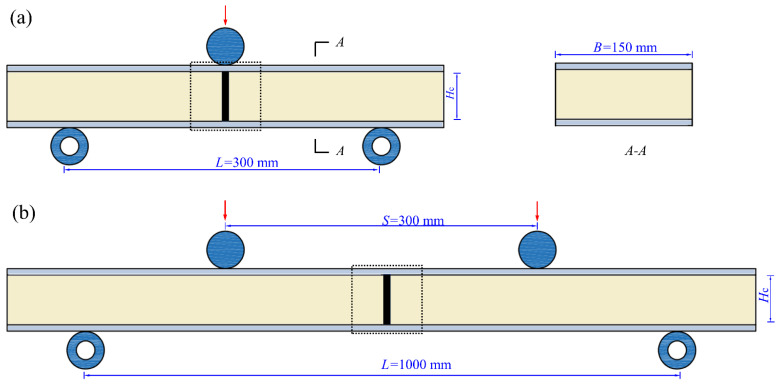
Schematic of panel-to-panel construction for three-point (**a**) and four-point (**b**) bending.

**Figure 6 materials-14-05099-f006:**
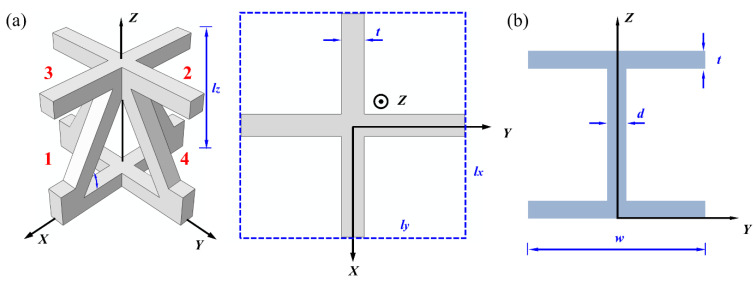
Schematic of the representative unit cell (**a**) and I-beam element (**b**).

**Figure 7 materials-14-05099-f007:**
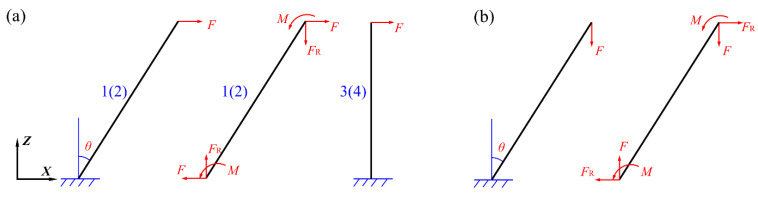
Sketch of loading distribution on individual truss under shearing (**a**) and compression (**b**).

**Figure 8 materials-14-05099-f008:**
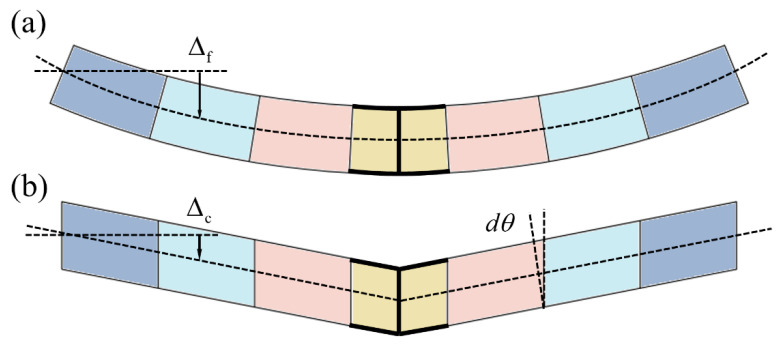
Deformation sketch of sheet metal (**a**) and lattice core (**b**).

**Figure 9 materials-14-05099-f009:**
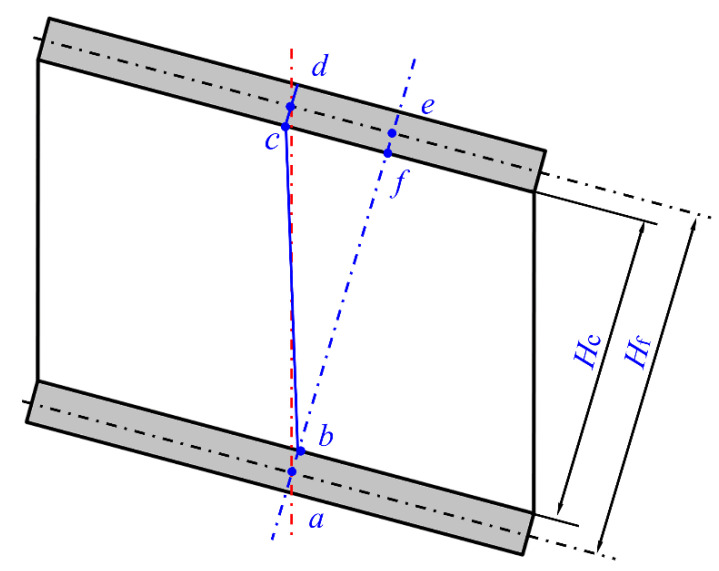
Shearing deformation sketch of lattice core.

**Figure 10 materials-14-05099-f010:**
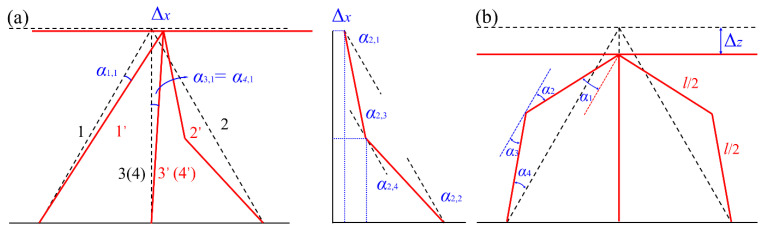
Sketch of the plastic deformation mode for the truss under shearing (**a**) and compressive (**b**) stress.

**Figure 11 materials-14-05099-f011:**
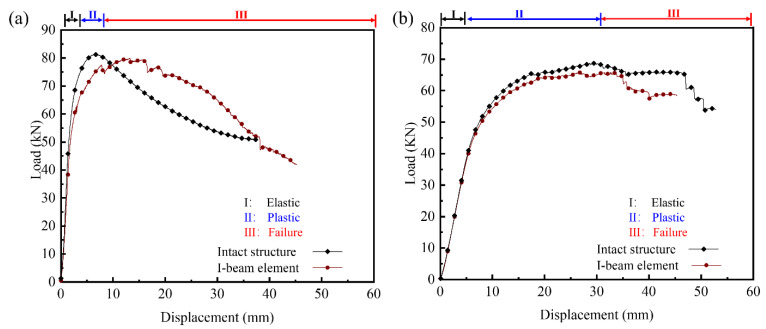
Experimental load-displacement curves for 2 types of panel-to-panel construction under three-point (**a**) and four-point (**b**) bending.

**Figure 12 materials-14-05099-f012:**
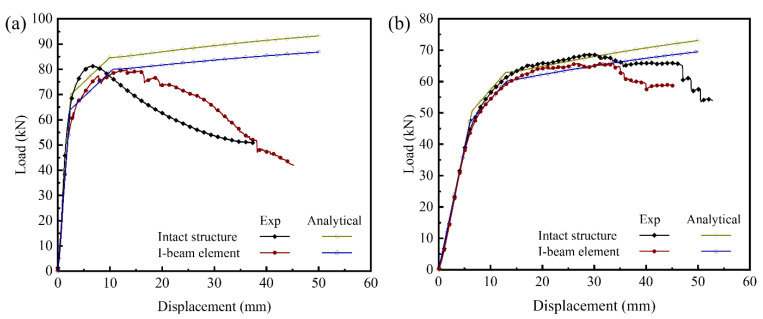
Load-displacement curves obtained from analytical model and experimental results under three-point bending (**a**) and four-point bending (**b**).

**Figure 13 materials-14-05099-f013:**
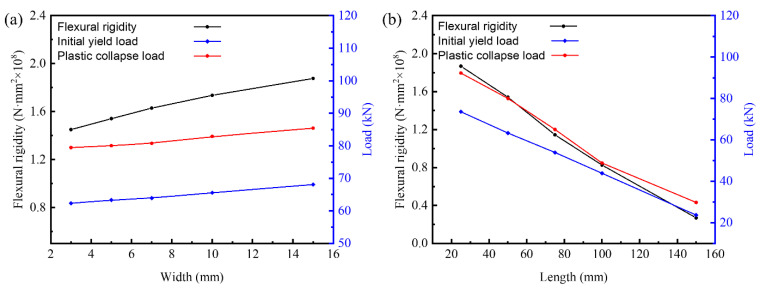
Effects of beam width (**a**) and length (**b**) on properties of the panel-to-panel construction.

**Figure 14 materials-14-05099-f014:**
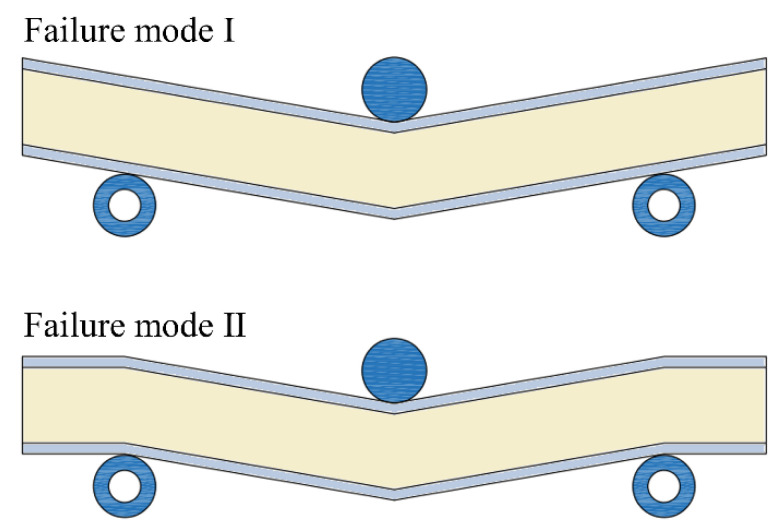
Schematic of deformation modes for the panel-to-panel construction under three-point bending.

**Table 1 materials-14-05099-t001:** Chemical composition of Q345 low alloy steel (%).

Element	Mn	Si	P	S	Cu	Ni	Cr	V	Ti	Fe
Content	0.75	0.45	0.025	0.020	0.05	0.02	0.03	0.002	0.2	Balance

**Table 2 materials-14-05099-t002:** Mechanical properties of base material and butt joint (MPa).

	Elastic Modulus	Yield Strength	Ultima Strength
Base material	207.3	345.13	498.27
	206.2	340.06	488.85
Butted joint	209.6	349.77	502.72
	208.3	345.11	505.46

## Data Availability

Data sharing not applicable.

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
