# Peer review of "Experimental and Analytical Analysis of Mechanical Properties for Large-Size Lattice Truss Panel Structure Including Role of Connected Structure"

_materials, 2021, doi:10.3390/ma14175099_

Round 1

Reviewer 1 Report

The reviewer comments of the paper «Experimental and Analytical Analysis of Mechanical Properties for Large-size Lattice Truss Panel Structure Including Role of Connected Structure»- Reviewer

The authors presented an article «Experimental and Analytical Analysis of Mechanical Properties for Large-size Lattice Truss Panel Structure Including Role of Connected Structure». However, there are several points in the article that require further explanation.

Comment 1:

The abstract needs to be improved.

Demonstrate in the abstract novelty, practical significance. Add quantitative and qualitative work results to the abstract.

Comment 2:

The introduction needs to be improved.

Firstly, group quotation is unacceptable in one phrase, for example [1-4], [8-10]. Break this sentence into parts or individual sentences. For example, ... [...], ... [...], etc. Or one reference - one sentence.

Now the list of references needs to be supplemented with at least 6-8 more references published over the past 3 years.

It is necessary to add a paragraph and a detailed analysis of the studied material of the constructions. Why is this material so important?

After analyzing the literature, show before formulating the goal of the "blank" spots. Which has not been previously done by other researchers. You must show the importance of the research being undertaken. Show what will be the new research approach in this article. You need to show a hypothesis.

Add a clear purpose to the article.

Comment 3:

  1. Experimental

Are all figures original? If not needed appropriate citations and permissions.

Add the physical and mechanical properties of the test material to the table. You need an appropriate reference.

Describe the measurement procedure in more detail. At what point in time? How is the measuring setup set up? How many repetitions of measurements? What statistical methods are used to process experimental results? Describe the experimental stand in more detail. What method of experiment planning is used and why?

Comment 4:

  1. Analytical

Are all formulas original? If not needed appropriate citations.

Comment 5:

It will be useful to add a section of Nomenclature in which to sign all the physical quantities and abbreviations encountered in the article. There are many physical quantities in the text and such a section will help to find the description of the necessary element.

For example,

  • : Density (kg/m3)

LTPS        : Lattice truss panel structure

etc.

Comment 6:

Conclusions.

It is necessary to more clearly show the novelty of the article and the advantages of the proposed method. Add qualitative and quantitative results of your work. What is the error of the obtained models? What is the difference from previous work in this area? Show practical relevance. Conclusions should reflect the purpose of the article.

Authors should more clearly justify the relevance of this article. What is its scientific novelty and practical significance? Authors should carefully study the comments and make improvements to the article step by step. After major changes can an article be considered for publication in the "Materials".

Author Response

We would like to thank for your positive and constructive comments and suggestions. The manuscript has been improved greatly, and the detailed responses are listed below.

1. The abstract needs to be improved. Demonstrate in the abstract novelty, practical significance. Add quantitative and qualitative work results to the abstract.

     Thanks very much for your good suggestion. In our revised abstract, we introduced “The large-size lattice truss panel structure (LTPS) is continually increasing for the higher upsizing, but the roles of its connected structures on the mechanical properties is always ignored during the previous structural integrity assessment” to demonstrate the importance and practical significance of our study. And the quantitative informations were also added as “the maximum error was limited within 20%.” and “the dimensional effects of the connection components on mechanical properties were also analyzed by the theoretical model, and indicated that the panel-to-panel LTPS will present better mechanical performance than the intact structure when the width of I-beam element exceeds 12.2mm or the its length downgrades to 39.1mm, which provide a comprehensive guidance for the engineering design of large-size LTPS”.

2.The introduction needs to be improved.

    (1) Firstly, group quotation is unacceptable in one phrase, for example [1-4], [8-10]. Break this sentence into parts or individual sentences. For example, ... [...], ... [...], etc. Or one reference - one sentence. Now the list of references needs to be supplemented with at least 6-8 more references published over the past 3 years.

(2) It is necessary to add a paragraph and a detailed analysis of the studied material of the constructions. Why is this material so important?

(3) After analyzing the literature, show before formulating the goal of the "blank" spots. Which has not been previously done by other researchers. You must show the importance of the research being undertaken. Show what will be the new research approach in this article. You need to show a hypothesis.

    Thanks very much for your good suggestion. The introduction has been revised as

    (1) The group quotation of the references has been deleted, and 7 references published at last 3 years were supplied, including Du et al., 2021; Kim et al., 2021; Wu et al., 2021; Du et al., 2020; Du et al., 2021; Wang et al., 2019; Qiao et al., 2020; Wang et al., 2020.

    (2) The relevant introduce for the studied material was added in Page 6.

    (3) The shortage of the previous study and the significance of the analytical model were summarized as “However, the above researches were conducted by the FEM, and the experimental verification was still absent, and its aim was to reveal the deformation mode, bearing capacity, ductility, degradation property and fatigue life of the joint. The effect of the connected geometric dimensions on the mechanical properties of the panel-to-panel LTPS is ambiguous. In addition, a theoretical model, which not only is more convenient for engineering application but also reflects more deeper understanding of deformation mechanism, is also still required to predict the overall deformation behavior of the panel-to-panel construction.” in page 5.

3. Are all figures original? If not needed appropriate citations and permissions. Add the physical and mechanical properties of the test material to the table. You need an appropriate reference.

    Thanks very much for your good comment. In our revised manuscript, all experimental figures and data are original. And the measured mechanical properties of the Q345 steel has been added, as shown in table 2 (Page 7).

    The measurement procedure in more detail. At what point in time? How is the measuring setup set up? How many repetitions of measurements? What statistical methods are used to process experimental results? Describe the experimental stand in more detail. What method of experiment planning is used and why?

    Thanks very much for your good suggestion. The experimental details including clamping, loading and measurement method were added as “The samples were cut with a parallel length of 240mm and a fillet of 25mm by a high-pressure water jet, as exhibited in Fig. 2(a). The upper chuck was fastened. But the lower chuck was imposed on a constant displacement with speed of 2.0mm/min until the specimen fracture. The deformation of the parallel section was monitored and controlled by an axial extensometer with a clip gauge of 20mm, and the magnitudes of axial loads imposed to the specimen were measured by a force sensor. To improve the accuracy, two groups of repeated tests were conducted” and “The deformation of the structure was monitored by an axial displacement extensometer with a gauge of 25mm. The loads imposed on the specimen were measured by a force sensor”. More detailed information was added in pages 6 and 10.

4. Are all formulas original? If not needed appropriate citations

    Thanks very much for your good comment. All equations involved in the manuscript were proposed based on the geometrical characteristics of pyramidal lattice truss panel and the equilibrium theory. And we make sure that they are acceptable easily by the readers according to our description without the citations.

5. It will be useful to add a section of Nomenclature in which to sign all the physical quantities and abbreviations encountered in the article.

    Thanks very much for your good suggestion. In our revised manuscript, the Nomenclature has been added in page 2.

6. It is necessary to more clearly show the novelty of the article and the advantages of the proposed method. Add qualitative and quantitative results of your work. What is the error of the obtained models? What is the difference from previous work in this area? Show practical relevance. Conclusions should reflect the purpose of the article.

    Thanks very much for your good suggestion. In our revised Conclusion part, we have addressed that the roles of connected structure which is always ignored by the previous researches were revealed extensively in our study, to provide a guidance for the engineering application and design of the large-size LTPS. And the qualitative experimental and theoretical results were analyzed, and the error of the prediction was calculated and explained in pages 30 and 31.

Reviewer 2 Report

The paper presents an experimental and analytical study of steel trussed lattice panels, with an aim to describe the role of the connecting element (an I-beam). Overall, both the experimental and analytical work are not adequately described to support the analysis and conclusions of the paper, thus the conclusions cannot be considered as robust and reliable. Specifically:
1. these panels would have deformed in a complicated manner, and indeed the authors allude to 3 stages of elastic then plastic deformation, bending and shearing, fracture and buckling and debonding of the panel. These mechanisms have not been adequately described for the reader to understand the process of panel deformation and failure, in neither the intact nor connected panels
2. the analytical modelling is described similarly, where the various complicated mechanisms are not explained adequately, nor adequately shown how these reflect the mechanisms seen in the experiments. Also, the variables are poorly described, and at times new and unexplained variables appear in equations as they progress
3. it is not clear how the analytical model is used to create the tri-linear curves in Fig 12, nor why the “plastic collapse stage” is actually increasing in load resistance
4. Fig 11 – which is “experimental” and which is “simulated”?
5. Fig 12 – which is “experimental”, “analytical” and “FEM”? (FEM was not mentioned previously)
6. the dimensional analysis of the I-beam is inadequately described thus the conclusions regarding the role of this connecting element are not reliable

Author Response

        We would like to thank for your positive and constructive comments and suggestions. The manuscript has been improved greatly, and the detailed responses are listed below.

  1. These panels would have deformed in a complicated manner, and indeed the authors allude to 3 stages of elastic then plastic deformation, bending and shearing, fracture and buckling and debonding of the panel. These mechanisms have not been adequately described for the reader to understand the process of panel deformation and failure, in neither the intact nor connected panels.

        Thanks very much for your good comment. In our revised manuscript, we gave a more extensive introduction about the 3-stage deformation processes in pages 21-23. The relevant parameters of the mechanical properties, including flexural rigidity, initial yield and plastic collapse load, were analyzed qualitatively. The downgrade effect of the connection element (I-beam) was explained. Based on the experiments, in stage I: the overall deformation was dominant by the elastic bending deformation of the metal sheets and coerced shearing deformation of the inner core; in stage II: the inner core remains elastic deformation, but the metal sheets yield; in stage III: inner core occurs plastic deformation.

  1. The analytical modelling is described similarly, where the various complicated mechanisms are not explained adequately, nor adequately shown how these reflect the mechanisms seen in the experiments. Also, the variables are poorly described, and at times new and unexplained variables appear in equations as they progress.

        Thanks very much for your good comment. In our revised manuscript, to make the analytical modelling more clear and understandable, we have added several equations (Eqs. 6, 7, 50-57) and given more derailed introduction about the relationship between analytical equations. In addition, we checked whole manuscript to make sure all involved variables have been explained. The main process for the construction of analytical modelling in this manuscript can be concluded briefly as “Based on the Euler–Bernoulli beam theory, solid truss is regarded as engineering beam loaded in compression to calculated the deflection. Firstly, the longitudinal force N, shear force Q and moment M were calculated theoretically. Secondly, according to the boundary condition and moment balance, the equivalent shearing and elastic modulus models of the unit cell were obtained. Thirdly, based on the equivalent homogenization theory, the equivalent shearing and elastic modulus of the panel-to-panel LTPS (include I-beam) were derived. Finally, the flexural rigidity and elastic deformation equation were obtained. But the truss provides “hinged boundary” when the plastic hinge is formed, which arises bending deformation distinctly. The truss raises symmetric deformation on perpendicular bisector when the panel is subjected to compressive load. According to the energy balance, the work from the loading force is equal to the work absorbed by the plastic hinge, and the relation between loading force and displacement was obtained. For the whole structure, the similar energy method was used.”

  1. it is not clear how the analytical model is used to create the tri-linear curves in Fig 12, nor why the “plastic collapse stage” is actually increasing in load resistance

        Thanks very much for your good comment. In our revised manuscript, we have added several equations to introduce extensively the production of predicted cures in Fig. 12 in pages 24 and 25.

  1. Fig 11 – which is “experimental” and which is “simulated”?

    Thanks very much for your good comment. The title of the Fig. 11 has been improved as “Experimental load-displacement curves for 2 types of panel-to-panel construction under three-point (a) and four-point (b) bending” in page 23.

  1. Fig 12 – which is “experimental”, “analytical” and “FEM”? (FEM was not mentioned previously)

      Thanks very much for your good comment. The title of the Fig. 12 has been improved as “Load-displacement curves obtained from analytical model and experimental results under three-point bending (a) and four-point bending (b)” in page 26.

  1. The dimensional analysis of the I-beam is inadequately described thus the conclusions regarding the role of this connecting element are not reliable

       Thanks very much for your good comment. In our revised manuscript, to make the analysis and conclusion more reliable, on the one hand, the quantitative effect of the I-beam width d and length w on the mechanical properties (flexural rigidity, initial yield and plastic collapse load) of the panel-to-panel LTPS were analyzed and concluded, and the downgrade effect of the connection element (I-beam) was explained. Based on the analytical models, the alternative proposals of the were suggested. On the other hand, a corresponding qualitative theoretical explanation was also added in pages 27, 28.

Reviewer 3 Report

The manuscript "Experimental and Analytical Analysis of Mechanical Properties for Large-size Lattice Truss Panel Structure Including Role of Connected Structure" has been reviewed. The subject investigated is interesting. However the presentation way and the flow of information presented in the manuscript is confusing for the reader. It is difficult to understand the objectives of this work. Also the title should be revised.

Results and discusssion should be separated paragraphs. Results and discussion are not supported by experimental results.

In conclusions the authors have to majorly revise the manuscript before reconsidering for the review and further process.

Author Response

    We would like to thank for your positive and constructive comments and suggestions. The manuscript has been improved greatly, and the detailed responses are listed below.

  1. It is difficult to understand the objectives of this work. Also the title should be revised.

    Thanks very much for your good comment. In our revised manuscript, to make the objective of this study more clear, we have improved the Introduction part to address that “With the development of industry, the demand for the large-size LPTS is continually increasing for the higher upsizing. But at present, most existing researches are performed based on a representative element for simplicity, as the role of connected structures on the mechanical properties of the large-size LPTS is ignored.” In page 3. Thus, we emphasized the “Large” and “Role of Connected Structure” in the title. In addition, in page 5, the shortage of the previous study and the significance of the analytical model were summarized as “However, the above researches were conducted by the FEM, and the experimental verification was still absent, and its aim was to reveal the deformation mode, bearing capacity, ductility, degradation property and fatigue life of the joint. The effect of the connected geometric dimensions on the mechanical properties of the panel-to-panel LTPS is ambiguous. In addition, a theoretical model, which not only is more convenient for engineering application but also reflects more deeper understanding of deformation mechanism, is also still required to predict the overall deformation behavior of the panel-to-panel construction.”

  1. Results and discussion should be separated paragraphs. Results and discussion are not supported by experimental results.

    Thanks very much for your good comment. In our research, we paied more attention on the construction of analytical model of large-size lattice truss panel structure including role of connected structure due to its more convenient engineering application and deep deformation mechanism, as introduced in the section 3-“Analytical”. And the experiments were employed to verify the accuracy of the proposed model. Thus, the combination of results and discussion is more desire for the better understanding of analytical model. But in revised manuscript, we added more quantitative information of experimental results and comparison between experimental and predicted results. The dimensional effect of the I-beam width d and length w on the mechanical properties (flexural rigidity, initial yield and plastic collapse load) was also improved greatly for a more reliable conclusion.

Round 2

Reviewer 1 Report

The authors have improved the article according to the comments. The article can now be published.

Reviewer 2 Report

The authors have substantially improved the paper. 

Reviewer 3 Report

Accept in present form